# Chagas Disease in Latin America and the United States: Factors Influencing Differences in Transmission Rates Among Differing Populations and Vectors [note 1]

**DOI:** 10.3390/insects16060570

**Published:** 2025-05-28

**Authors:** Stephen A. Klotz

**Affiliations:** Division of Infectious Diseases, University of Arizona College of Medicine, 1501 N. Campbell Ave., Tucson, AZ 85724, USA; sklotz@u.arizona.edu; Tel.: +1-520-429-9017

**Keywords:** Chagas disease, triatomes, *Trypanosoma cruzi*, kissing bugs, fecal–oral contamination

## Abstract

Chagas disease is rare in the United States and common in Latin America even though both have insect vectors that bite home residents and mammalian reservoirs of the parasite, *Trypanosoma cruzi*. The commonly accepted mode of vectorial transmission of the parasite is admittedly improbable. Therefore, this paper argues for a wider acceptance of oral or fecal-oral transmission of Chagas disease in both geographic areas. The likely differences in the rates of disease in the two areas are discussed.

## 1. Introduction

*Trypanosoma cruzi*, a protozoan parasite that causes Chagas disease, is found in kissing bugs, small mammals and humans throughout much of Latin America. In the past, millions of human Chagas disease cases occurred, and they continue to occur in smaller numbers presently. Although *T. cruzi* is enzootic in mammals and kissing bugs within the lower two-thirds of the United States (US) [1], it is rarely transmitted to humans. One is prompted to ask why the transmission rate of *T. cruzi* to humans is different in the US compared with Latin American countries. This report addresses the many factors which together may explain the differences. Understanding these differences may help determine what prevention strategies are important.

Most authorities accept that the transmission of *T. cruzi* to humans occurs by the so-called vectorial mode. Definitions of this mode are provided in several examples. Up to Date, a popular online resource for physicians, states, “Humans usually become infected when the triatomine vector defecates during its blood meal and fecal material containing the parasite is inoculated through the bite wound or intact mucous membranes” [2]. Kirchoff is more detailed in his description: “When infected insects defecate during subsequent blood meals they often deposit parasite-laden faeces that can result in transmission by coming into contact with vulnerable surfaces such as the conjunctivas, nasal and oral mucosas, or the bug bite itself…” [3]. It is clear that this mode of transmission makes the assumption that an infected kissing bug defecates shortly after feeding on or near the bite site, and the victim rubs feces onto mucus membranes or the bite wound. This hypothesis has prevailed as the explanation for infection over the course of the 20th century in Latin America. Although autochthonous Chagas disease was not recognized in the US until 1955 [4], the vectorial hypothesis, by default, has been accepted as the mode of acquiring Chagas disease in the US even though the species of kissing bugs involved in human disease are numerous and their geographical distributions also various.

## 2. Chagas Disease Transmission: Vectorial Versus Fecal–Oral Spread

### 2.1. Discoveries in Latin America

Carlos Chagas, working in a small laboratory on a railcar in Minas Gerais, Brazil, discovered the parasite, the intermediate insect host and the disease bearing his name all within the span of several years [5,6,7]. However, some of Chagas’ conclusions were mistaken. For example, goiter was a common problem among the people of Minas Gerais, and Chagas believed that it was caused by infection with *T. cruzi* (rather than iodine deficiency). Chagas learned of kissing bugs, or “barbieros”, from railroad workers who complained to him of the nocturnal bloodsucking bugs. Fully aware that malaria involved hematophagous insects, Chagas reasoned that *T*. *cruzi*, like malaria, must be transmitted via saliva during the bite [8]. This, of course, was incorrect, and the error was compounded by findings obtained by Oswaldo Cruz, Chagas’ mentor. Cruz demonstrated that kissing bugs were vectors of *T. cruzi* by placing infected kissing bugs (sent to him by Chagas) with caged marmosets, some of which became infected with *T*. *cruzi.* Chagas (and Cruz) believed that the parasite was transmitted during the bite by the insect, overlooking the more likely explanation, i.e., that the marmosets ate the insects and thus were infected by the oral route [9].

### 2.2. Discoveries in the Southwest US

Kofoid and Mcculloch in 1916 reported *T. triatomae (T. cruzi)* in the gastrointestinal tract of *Triatoma protracta* that were found in woodrat nests in the Southwest [10]. The bugs were feeding at night on prospectors sleeping in the desert. Later, Kofoid and Donat, aware of Chagas’ claim that the bite was the source of infection, tried to prove it but were unable to do so. They allowed kissing bugs to feed to repletion on mice, but no mouse became infected [11]. Kofoid and Donat were aware that Brumpt had demonstrated the maturation of *T. cruzi* within the gastrointestinal tract of kissing bugs, bed bugs and a soft-bodied tick [12], and the two investigators thereafter confirmed that *T. cruzi* was passed in the feces of kissing bugs, not saliva [11].

Sherwin Wood, working in the Southwest, seized upon the importance of kissing bug feces in the pathogenesis of Chagas disease and attempted to quantify kissing bug defecation [13]. He remarked that in the laboratory, kissing bugs do not remain in the vicinity of the host animal after feeding. Defecation by *T. protracta* sometimes occurred hours after feeding, and some bugs refused to defecate at all [13]. Kissing bug behavior of the very kind described by Wood was captured on film by a photographer being fed upon by *T. rubida* [14].

However, if the vectorial hypothesis regarding the transmission of Chagas disease is correct, then rapid and frequent defecations around the bite site would be important. Thus, Zeledon, after observing the feeding and defecation of *Rhodnius prolixus*, *Triatoma infestans* and *Triatoma dimidiata* (three Latin American kissing bugs), proposed a defecation index in 1977 [15]. This index is now *de rigueur* for any investigator working on kissing bugs, even though it has little in common with the natural world. The index is obtained by observing kissing bugs fed warmed blood from a container, often a condom. To keep the blood fluid and to prevent clotting, an anti-coagulant (citrate) must be added to the blood. The anti-coagulant likely hastens bug defecation/urination since the food bolus cannot coagulate. Experience has shown that citrated blood, furthermore, is not optimal for kissing bug maturation, as bugs are unable to reach adulthood when fed exclusively citrated blood (personal communication, C. Reisenman).

Similar to Wood’s findings [13], we found that three US kissing bug species (*Triatoma protracta*, *T. rubida* and *T. recurva*) were reluctant to defecate on or near the animal host [16]. Furthermore, in a survey of kissing bug bite victims involving thousands of bites, we found that the same three species did not bite near the eyes and mouths of humans (n = 105) with any great frequency and no victim contracted Chagas. Bugs preferred biting the limbs and torso; consequently, the likelihood of kissing bug feces contacting victims’ mucous membranes was low [17]. Nevertheless, an elderly Louisianan woman (with frequent travels to Latin America) tested positive by serology and xenodiagnoses for presumed autochthonous Chagas had fecal smears from *T. sanguisuga* found in her home on her nightgown [18].

Following the numerous reports of oral Chagas, one might ask if the vectorial mode is still relevant in Latin America. It undoubtedly occurs, although it is estimated that infection due to this mechanism occurs once in ~1724 contacts with an infected bug [19]! The authors of this calculation commented thusly: “…transmission requires an extraordinary combination of somewhat unlikely events. An infected vector has to defecate sufficiently close to the biting site whilst or shortly after feeding, the infected feces must be brought to the bite wound by the host by scratching, and the pathogen then has to cross the skin of the host to initiate infection” [19].

## 3. More About the Modes of Chagas Transmission

Fecal–oral spread is the likely mode of transmission among wild mammal reservoirs of *T. cruzi* [9]. A common feature of these mammals is that they nest in burrows or trees where they may eat live bugs living in their nests and swallow feces adherent to their fur in the process of grooming. Insects of all types are ingested by small mammals. For example, the stomach contents of wild raccoons and opossums taken from natural surroundings in the US were characterized, and insects constituted 48% and 21% of the food sources, respectively [20]. It is noteworthy that Chagas is enzootic in both these mammals in the US [21].

Data from past studies by Chagas are undoubtedly skewed toward “vectorial transmission”, as it was a widely held consensus among authorities that vectorial transmission was the predominant means of transmission. However, kissing bug fecal contamination of human food and drink has recently gained notoriety. This mode of disease acquisition undoubtedly occurred in the past but likely went unrecognized. There are now many reports of fecal–oral outbreaks involving large numbers of individuals in Latin America. For example, from 2000 to 2013, there were 1570 acute cases of Chagas in northern Brazil, of which 68.9% were due to oral transmission and 6.4% were attributed to vectorial transmission [22,23]. These oral outbreaks are often confined to certain ecotypes, especially forested areas inhabited by tree-dwelling bugs such as *Rhodnius* species [23].

Traditionally, victims of vectorial transmission presented with facial swelling or Romana’s sign (considered proof that fecal matter was rubbed onto the mucous membranes following a bite). This sign is rarely seen, as most acute cases of Chagas go undiscovered [24]. Furthermore, the sign is not pathognomonic of Chagas disease. It is seen with such parasitic illnesses as leishmaniasis, trichinellosis and outbreaks of oral Chagas disease and even occurs as an allergic reaction to bites by uninfected kissing bugs [25]!

Historically, Chagas disease in Latin America was recognized predominately among residents of single-roomed dwellings with wattle and daub walls and palm-thatched roofing. There, “…intradomiciliary colonies [of triatomes] with average densities above 3000 insects, exceptionally around 10,000, [were] maintained by the food sources found within the homes, especially humans, dogs, cats, rodents and other animals that sleep indoors” [26]. One can easily appreciate that in such a home, surfaces and occupants would be contaminated with bug feces containing viable *T. cruzi.* Indeed, Zeledon remarked in one report that feces fell from the ceiling of human dwellings, possibly into the eyes and mouth of the occupants [27].

It is, therefore, more than possible that some and maybe all autochthonous human Chagas cases in the U.S. occurred by fecal–oral transmission, perhaps from fecal contamination of food and drink [23]. Many variations of fecal transmission may occur. For example, I was sent an email from Tucson, Arizona, with a photograph of a live *T. rubida* walking on the tonsil of an institutionalized elderly woman (photographed by the daughter). From another long-term care institution, I received a dead *T. rubida* that was lodged within an elderly woman’s swollen buccal membranes for several weeks [28]. The survival of *T. cruzi* from recently sprayed dead vectors may last up to a month [29]. In that report dead *T. infestans* were found on clothes, furniture, and food in the homes, and the authors speculated that fecal material may be transferred from inert surfaces and fomites to humans [29]. It is also noteworthy that infection could be spread by such wild mammals as opossum [30,31,32]. Infection could also spread during occupational activities, as has been hypothesized for palm fiber workers [33], or during recreational activities such as hunting [34].

### 3.1. Does It Matter What Mode of Transmission Occurs?

If we continue to exclusively invoke vectorial transmission in the US as the route of infection the public will focus entirely on the act of biting with its remote possibility of infection as explained before. Being bitten by a kissing bug is more likely to lead to accelerated allergic reactions, including anaphylaxis, rather than infection and has led to several deaths [34]. Indeed, kissing bugs are the most common cause of anaphylaxis among biting insects in the USA [34].

Annually, in the foothills of Tucson, AZ, there are many hundreds (perhaps thousands) of kissing bug bites. Each May, June and July, I receive requests for Chagas serology by individuals bitten by kissing bugs. Given the unlikelihood of a bite leading to infection, it is perhaps more important to convince bite victims to eliminate kissing bugs from their homes. Kissing bugs are encountered almost exclusively in the home setting, yet most bite victims are unaware of factors that favor kissing bug home intrusion and persistence (Table 1). Many homeowners are unaware of inexpensive approaches to lessening kissing bug contact, such as removing clutter in and around the house and removing bug refugia such as wood, lumber, trash and rodent nests [35]. For example, in conducting home surveys in southern Arizona, we noted that homeowners kept chickens in coops that adjoined the home. The coops were colonized by kissing bugs that fed on the homeowners and their children in addition to the chickens. We found that pet dogs and cats also served as blood meals for the bugs. If homeowners refuse to prohibit the pets from entering the home, then the pets should at least be prohibited entry to the bedroom, where bugs hide during the day. Food and drink should not be left uncovered overnight. We have found dead kissing bugs in drinking glasses in kitchens. It is possible that attention to house hygiene may prevent some autochthonous Chagas cases in the US. Using home repair measures focusing on the walls and ceilings have been implemented with good results in Latin America [36].

### 3.2. What Are the Major Factors Accounting for the Differences in the Transmission of Chagas Disease in Latin America and the US?

Autochthonous cases of Chagas disease in humans in the US are rare, notwithstanding the estimate of 10,000 autochthonous cases [37]. Historical and geographical preferences in home construction and rare domiciliation of the home by kissing bugs are likely the deciding factors contributing to the low transmission of Chagas disease in the US compared with Latin America. Cultural forces have been very important in shaping the epidemic in Latin America, especially in rural isolated homes (families) and communities. Many of these communities consist of impoverished people with a subsistence living and few hygienic resources (see Table 1). Together with numerous kissing bug species capable of domiciliation (e.g., *T. infestans* or *Rhodnius prolixus*) the stage is set for generations of new infections. There are many poor people in substandard housing in the US, but the indigenous kissing bug species demonstrate no predilection for domiciliation, even in the Southwest, where bugs seasonally enter homes. The genetics of the various human populations likely play a role in the susceptibility to and severity of Chagas Disease, but this has not been elucidated.

## Figures and Tables

**Table 1 insects-16-00570-t001:** Factors favoring transmission of Chagas disease to humans (Latin America vs. US).

Home Characteristics That Affect Kissing Bug Residence in Homes	Latin America	United States
Kissing bug entry into homes	Almost impossible to prevent	Almost impossible to prevent; common happenstance in the desert Southwest
Peridomestic environments with potential blood meal hosts	In rural areas, livestock are commonly kept close to the house	Cats and dogs are commonly kept inside the home; chickens are kept in a coop adjacent to the home
Domiciliation of kissing bugs in homes	Continues to occur especially in homes with palm leaf roofing or wattle and daub walls	Probably does occur but rarely; in the desert Southwest, it is a seasonal event, i.e., intrusion into the home and biting continues until homeowners recognize the problem
Homes comprised of one multipurpose room, i.e., for sleeping, eating and socializing	Formerly a commonplace in dwellings in impoverished rural settings	Rare in the US
Temperature control in the home space	Rare	A norm in US home construction
Dwellings without doors and screens	Historically commonplace in some areas of Latin America	US homes typically possess doors and screens
Application of insecticides in home space	Historically, not a common occurrence	Insecticides for use within the home are readily available to homeowners, and there is a thriving commercial pest control industry

## Data Availability

The original contributions presented in this study are included in the article. Further inquiries can be directed to the corresponding author.

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
