# Peer review of "Chagas Disease in Latin America and the United States: Factors Influencing Differences in Transmission Rates Among Differing Populations and Vectorsâ€"

_insects, 2025, doi:10.3390/insects16060570_

Round 1
Reviewer 1 Report
Comments and Suggestions for Authors
Insects-3545160 Review
This manuscript brings an interesting discussion on the Chagas disease transmission in Latin America and in USA. The main idea of the manuscript is to compare, over time, the cases transmitted by insect vectors and the cases by oral transmission. Despite the interesting objective presented by the author, the contextualization, the technical comments and terms, do not support the complexity of the subject presented. In addition to that, this manuscript presents several conceptual mistakes and misinformation detected throughout the text. Therefore, I regret to inform this manuscript can not be recommended for publication in the Insects Journal.
A few mistakes found in the text:
Title is too broad
Introduction is not well contextualized.
“Most authorities believe the transmission of T. cruzi to humans occurs by the so called vectorial mode”
R. prolixus is mentioned as a possible vector responsible for Chagas disease cases transmitted orally in Brazil. The mentioned species is not reported as distributed in that country.
“T. cruzi is endemic in mammals”. The term endemic is not applicable in this sense
In summary, the Chagas disease is a very complex matter in all of its aspects. Here is listed some of its complexities that must be taken into account: the variety of symptoms and clinical cases; the distinct clinical aspects occurring in distinct geographical areas; the transmission aspects; the distinct more them 150 vector species presenting distinct ecology; some species vectors are highly infected in some areas while in other areas they are not; some species vectors tend to defecate immediately and near the food source while others don´t; there is not a behavioral pattern for all species vectors or even the same species might behavior in different ways according to the ecological aspects in the related geographical region.
The author presents references that support his objectives however, the complexity of the subject is far beyond the presented arguments.
The subject of the manuscript is relevant but the arguments and contextualization is not in accordance with the complexity of the topic.
Author Response
Response: I thank the Reviewer for taking the time to read the manuscript. I appreciate the comments and have attempted to correct what objections I could and hone the argument a little finer.
Title is too broad. Response: This has been changed.
Introduction is not well contextualized. Response: I am not sure what I am to change or what is required here.
“Most authorities believe the transmission of T. cruzi to humans occurs by the so called vectorial mode” Response: believe has been changed to accepted.
R. prolixus is mentioned as a possible vector responsible for Chagas disease cases transmitted orally in Brazil. The mentioned species is not reported as distributed in that country. Response: This was changed to Rhodnius species
“T. cruzi is endemic in mammals”. The term endemic is not applicable in this sense Response: changed to enzootic.
In summary, the Chagas disease is a very complex matter in all of its aspects...Response: I entirely agree with the comments and tried to include some of these complexities in the edited manuscript
The author presents references that support his objectives however, the complexity of the subject is far beyond the presented arguments. Response: Hopefully with the edits made in the revision I have answered some of the Reviewer's objections.
Reviewer 2 Report
Comments and Suggestions for Authors
The author dissects the transmission of Chagas disease in the Americas using a comparison in the prevalence of the disease in Latin America and the USA.
This is very good work and this commentary is very important because it sheds light on the possibility that vector transmission of T. cruzi may not be as important as it is often taken to be. This commentary further explain the importance of fecal-oral transmission rendering the triatomes as reservoirs instead.
I highly commend the author for this great scientific piece.
Author Response
The author dissects the transmission of Chagas disease in the Americas using a comparison in the prevalence of the disease in Latin America and the USA. Response: Thank you for your kind comments. I sincerely believe we need to revisit some aspects of the "dogma" surrounding the transmission of Chagas. Rare presentations of infections such as Romana's sign are perhaps self explanatory as to the mode of infection but details concerning the vast majority of infections are entirely unknown which is especially true for so-called autochthonous cases in the US. A positive serology provides no information on when, where or how the infection occurred.
Reviewer 3 Report
Comments and Suggestions for Authors
Please see file below for comments

Author Response
General Response: Thank you for your time and the detailed review of this manuscript. I have adopted almost all of your edits and the paper is stronger for it.
General Comments
Consider changing the article title. Response: It is changed to: Chagas Disease in Latin America and the United States. Why the transmission rates are so different for these different cultures, peoples and bugs?
The order of manuscript sections is confusing. Response: The order remains the same but some of the titles were changed. The section titles flow from the topic of the prior text.
Specific comments
1. The author claims...the author should mention the species of triatomines...Response: Triatomine species names are now included.
2. Vector species must be mentioned in the text...Response: Species names have been added to line 87 and 96.
3. The author appears to have a bias toward vectorial transmission...Response: I don't believe I have a bias; what I am trying to show is that invocation of the vectorial mode of transmission seems to be reflexive and ignores the many ways the pathogen could be transmitted by fecal-oral contamination.
Recommendations:
In line 103 it should be stated...Response: It is now clearly stated that this is an estimate.
It is essential...Response: I appreciate the importance of the epidemiology of the disease but it is not the major interest of this manuscript; I recently published a paper dealing with epidemiology and the concerns the Reviewer raises: Klotz SA. Epidemiology of Chagas Disease in the United States of America: A Short Review and Some Comments. Pathogens. 2025 Jan 1;14(1):24. doi: 10.3390/pathogens14010024. PMID: 39860985; PMCID: PMC11768169.
Minor points:
Keywords after Abstract and Introduction. Response: This was done.
Line 27. State T. cruzi is a protozoan. Response: This was done.
Line 28. Revise the sentence...Response: The sentence was changed.
Line35. Please consider revising the sentence... Response: the wording was changed to most authorities "accept"... You are correct that there is a broad consensus among those in the field but it is my opinion that most statements touting vectorial transmission are made reflexively, i.e., it is the default position as the mode of infection is entirely unknown in the majority of cases.
Round 2
Reviewer 1 Report
Comments and Suggestions for Authors
The manuscript was significantly improved
Author Response
Response: I have made some changes to the manuscript that are minor and aid in the reading of the text. Also changed the title to that suggested by another Reviewer.
Thank you for your suggestions.
Reviewer 3 Report
Comments and Suggestions for Authors
The author responded to the points I raised, and the resubmitted manuscript shows significant improvement. However, several issues still require minor revision before publication:
1.Title
The proposed title has strengths, such as clearly indicating the topic (Chagas Disease) and geographic scope (Latin America and the United States). However, the second part—"Why are the transmission rates so different for these different cultures, peoples and bugs?"—is somewhat lengthy and informal.
Using "cultures" to refer to Latin America and the U.S. is understandable but imprecise in a scientific context, as it typically relates to social practices or ethnic groups rather than epidemiological factors. Additionally, each country within Latin America has its own distinct “cultures.” The term "peoples" is technically correct but less common in contemporary scientific writing.
I suggest a more precise and formal title: "Chagas Disease in Latin America and the United States: Factors Influencing Differences in Transmission Rates Among Populations and Vectors"
2.Triatomine Species
I recommend adding a sentence at the end of the Introduction clearly stating that the vector species prevalent in Latin American countries differ from those in the U.S. Providing specific examples of these species would be very relevant, especially for an article published in Insects.
Minor points:
-Line 91: Triatoma dimidiata (revise species name)
-Reference 36: Correct the formatting of the citation.
Author Response
Comment: I suggest a more precise and formal title: "Chagas Disease in Latin America and the United States: Factors Influencing Differences in Transmission Rates Among Populations and Vectors" Response: Thank you for the suggestion; I have adopted your title
2.Triatomine Species
I recommend adding a sentence at the end of the Introduction clearly stating that the vector species prevalent in Latin American countries differ from those in the U.S. Providing specific examples of these species would be very relevant, especially for an article published in Insects. Response: This sentence has been added.
Minor points:
-Line 91: Triatoma dimidiata (revise species name): Response: The spelling has been corrected.
-Reference 36: Correct the formatting of the citation. Response: I cannot seem to correct the problem--this will need to be done by Editor.